# Supramolecular catalysis with ethers enabled by dual chalcogen bonding activation

Zhiguo Zhao[1], Yuanling Pang[1], Ziqiang Zhao[1], Pan-Pan Zhou[2] & Yao Wang [1]✉

The activation of ethers by weak interactions is a long-standing objective in supramolecular catalysis, but yet it remains an underdeveloped topic. The obstacles towards solving this problem are prominent since it is difficult for a weak interaction to cleave a relatively strong C-O σ-bond and moreover, the ionic intermediate composing of an alkoxide ion and an electrophilic carbocation would deactivate weak interaction donors. Herein, we describe a distinctive activation mode, dual Se⋯π and Se⋯O bonding, that could activate benzylic as well as allylic ether C-O σ-bonds to achieve cyclization, coupling and elimination reactions. This dual Se⋯π and Se⋯O bonding catalysis approach could tolerate various alkoxide leaving groups, while the other representative weak interaction donors showed no catalytic activity.

Catalysis with weak interactions has been established as a fundamental discipline in organic synthesis[1]. A large number of chemical transformations were achieved by hydrogen bonding interactions[2] as well as the lately established catalysis modes such as halogen[3–6], chalcogen[5–10] and carbon bonding[11] interactions. Generally, the success of this research area relies on the activation of comparatively reactive π–bonded functional groups, with carbonyl, imine, and nitro groups as dominant targets (Fig. 1a)[1]. It has been generally considered that it is difficult to weaken a strong σ–bond to generate reactivity by imposing of weak interactions on an organic molecule[1–12]. Accordingly, the catalytic activation of relatively strong σ–bonds by weak interactions has been recognized as a long-standing challenge in supramolecular catalysis[1–12]. In this context, research toward solving this severe limitation emerges as an urgent and essential goal.

As a representative case, even though benzylic as well as allylic ethers are frequently used reactants in organic synthesis, the activation of these ethers is not trivial. Generally, metal catalysts, superacid or strong photochemical conditions were employed to activate benzylic and allylic ethers[13–19]. Owing to the relatively strong C-O σ-bonds and the intractable alkoxide leaving group[20], it appears rather difficult for a metal-free approach to activate ethers, for instance, the coupling reactions between benzylic ethers and electron rich arenes requires the use of a stoichiometric amount of superacid BF$_3$–H$_2$O and

a high temperature (120 °C)[13]. In this context, the practices in supramolecular catalysis suggest the activation of ethers by weak interactions is an unfavorable approach[1–12], and it remains an underexplored topic (Fig. 1a). Apparently, the comparatively strong C–O σ–bond sets up a hard-to-reach barrier for weak interactions to play an effective role in activating ethers[12]. Moreover, even supposing that the cleavage of an ether C–O σ–bond by a weak interaction is possible, the ionic intermediate is constituted by two problematic species, a strongly alkaline and nucleophilic alkoxide and an electrophilic carbocation, which would potentially deactivate weak interaction donors by routine approaches such as competitive bonding, deprotonation, nucleophilic/electrophilic substitutions. Despite these glaring problems, however, the activation of ethers by weak interactions would serve as an important step for weak interactions to enter an area dealing with relatively strong σ-bonds.

Recently, contrary to the conventional covalent Lewis base property[21], disubstituted chalcogens that bear two lone pairs were used as electron acceptors to operate noncovalent chalcogen bonding interactions, which have been found applications in a broad chemical context including drug discovery, material chemistry, crystal engineering, anion recognition, and controlling of the conformations of reaction intermediates[22–31]. Several disubstituted chalcogens were used as chalcogen bonding catalysts, such as dithieno[3,2-b;2',3'-d]

[1]School of Chemistry and Chemical Engineering, Key Laboratory of the Colloid and Interface Chemistry of the Ministry of Education, Shandong University, Jinan 250100, China. [2]College of Chemistry and Chemical Engineering, Key Laboratory of Special Function Materials and Structure Design of Ministry of Education, Lanzhou University, Lanzhou 730000, China. ✉e-mail: yaowang@sdu.edu.cn

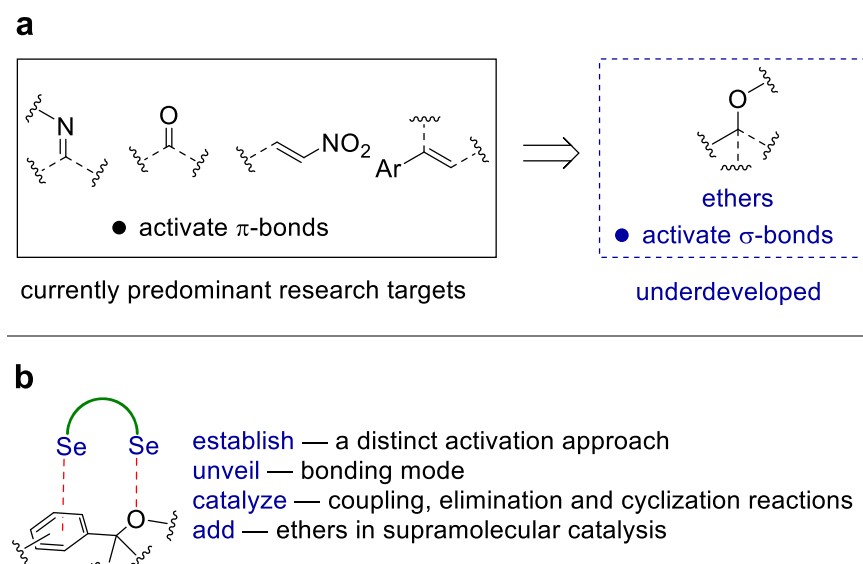

**Fig. 1 | Catalysis with weak interactions. a** An overview on catalysis with weak interactions. **b** This work: supramolecular catalysis with ethers.

thiophenes[32,33], bis(benzimidazolium) chalcogenides[34–38], and phosphonium chalcogenides[39–43]. To add ethers in the list of supramolecular catalysis, herein, we unveil a dual Se⋯π and Se⋯O bonding mode for the activation of ethers, and demonstrate its application in catalytic cyclization, coupling, elimination and the Friedel-Crafts reactions (Fig. 1b). Considering the substantial difference between the strong covalent activation and the weak interaction approaches, the dual chalcogen bonding catalysis approach has its unique advantage, which proceeds in the absence of any metal additive while it can give reactivity and enhance the reaction selectivity.

## Results
### Chalcogen bonding interactions

Initially, judging by the reported precedents in supramolecular catalysis[1], we were unaware that the weak strength of chalcogen bonding interactions would be capable of activating ether C-O σ-bonds. On the other hand, it was observed that the use of an ether solvent often resulted in an inhibition of reactions to some extent[41], indicating the presence of competitive chalcogen bonding between substrate and ether solvent. In this context, an investigation on chalcogen bonding interactions between selenides and ethers was carried out (Fig. 2). To this end, a representative chalcogen bonding donor **Ch1** was chosen as an optimal target to study chalcogen bonding interactions (Fig. 2a). Analysis of a 1:1 mixture of **Ch1** and ether **e1** in CD$_2$Cl$_2$ by $^{13}$C NMR revealed that there is almost no variation of the $^{13}$C signals of **e1** (for all carbons <0.05 ppm). Similar results were obtained using selenide **Ch2**, since almost no variation of the $^{13}$C signals of **e1** (for all carbons <0.05 ppm) was observed, even in the presence of a 2:1 mixture of **Ch2:e1**.

In contrast, however, analysis of a 1:1 mixture of **Ch3** and **e1** in CD$_2$Cl$_2$ by $^{13}$C NMR shows that the $^{13}$C signal of the carbon (C$_{Bn}$) attached to oxygen shifts downfield by 0.12 ppm. A much more dramatic variation of the chemical shift of $^{13}$C signals arises from the phenyl ring. The aromatic carbon C$_a$ shifts upfield as much as −0.48 ppm while the other aromatic carbons (C$_b$–C$_d$) shifts downfield from a range of 0.08 to 0.21 ppm. For comparison, the monodentate counterpart of **Ch3**, i.e. **Ch4**, was investigated. Analysis of a 2:1 mixture of **Ch4:e1** in CD$_2$Cl$_2$ by $^{13}$C NMR reveals that there is almost no perturbation of the $^{13}$C signals of **e1**.

To explain these experimental results, several possible bonding modes were depicted in Fig. 2b. The performance of monodentate **Ch2** and **Ch4** indicates a single interaction mode **COM1** or **COM2** is not productive. Meanwhile, the sharp contrast between **Ch3** and its

monodentate counterpart **Ch4** indicates a dual interaction mode is in operation. In addition, the variation of the $^{13}$C signals was observed not only from the carbon attached to oxygen of **e1** but more marked perturbation from the remote aromatic ring. These results suggest that a dual Se⋯π and Se⋯O bonding mode (**COM3**) is likely the operative one, while they are against mode **COM4**. In addition, since chalcogen bonding is almost a linear interaction[12], it is unfavorable to form **COM4** upon considering the bonding angles of **Ch3**.

To enable a dual Se⋯O and Se⋯π bonding mode **COM3**, a good match of the bonding sites between an ether and a chalcogen bonding donor is essential (Fig. 2c). The interaction of **Ch3** with different ether showed that the perturbation of the chemical shift of $^{13}$C signals of phenyl ring decreased dramatically upon increasing the distance between the oxygen atom and phenyl ring in ethers **e1**-**e3** (0.48, 0.23 and 0.14 ppm corresponding to C$_a$ of **e1**, **e2**, **e3**). Alternatively, upon maintaining ether **e1** unchanged, the increasing of the distance between two selenium atom leads to a decreased variation of the chemical shift of $^{13}$C signals. In the presence of **Ch5** and **Ch6**, only 0.09 and 0.05 ppm change of C$_a$ in **e1** were observed. Upon further increasing the chain length, bidentate **Ch7** performs like a monodentate selenide.

The steric hindrance arising from the substitutions of phenyl ring in a modified **e1** could block the Se⋯π interaction to some extent, thus resulting in a less marked variation of $^{13}$C signals of phenyl ring. To investigate Se⋯π interaction, three isopropyl groups were installed in the phenyl ring of **e1** to give a reference ether **e4**. The experimental result is in line with this notion, and a smaller variation (i.e. 0.24 vs 0.48 ppm) of the $^{13}$C signal of C$_a$ of **e4** was observed (Fig. 2d). In addition, an intramolecular competition experiment was carried out to corroborate the Se⋯π interaction. To this end, two aromatic rings with distinct steric hindrance were installed in ether **e5** (i.e. i-Pr$_3$Ph vs Ph). The experiment results showed that the $^{13}$C signal of C$_a$' in Ph varies 0.16 ppm, while a decreased perturbation of its steric counterpart C$_a$ in i-Pr$_3$Ph (−0.03 ppm) was obtained. In contrast, the variation of both the $^{13}$C signals of C$_a$' and C$_a$ in **e6** could be observed. These molecular control experiments further support a bonding mode of **COM3**.

The Se⋯O interaction would polarize the C-O bond to facilitate the heterolytic cleavage. On the other hand, the Se⋯π bonding leads to a distinct shift of the $^{13}$C signals of the aromatic ring. Using **e1** as an example, the $^{13}$C signal of aromatic carbon C$_a$ attached to benzylic carbon shifts upfield dramatically (−0.48 ppm), while the $^{13}$C signals of the other aromatic carbons (C$_b$–C$_d$) shift downfield (+0.08 to

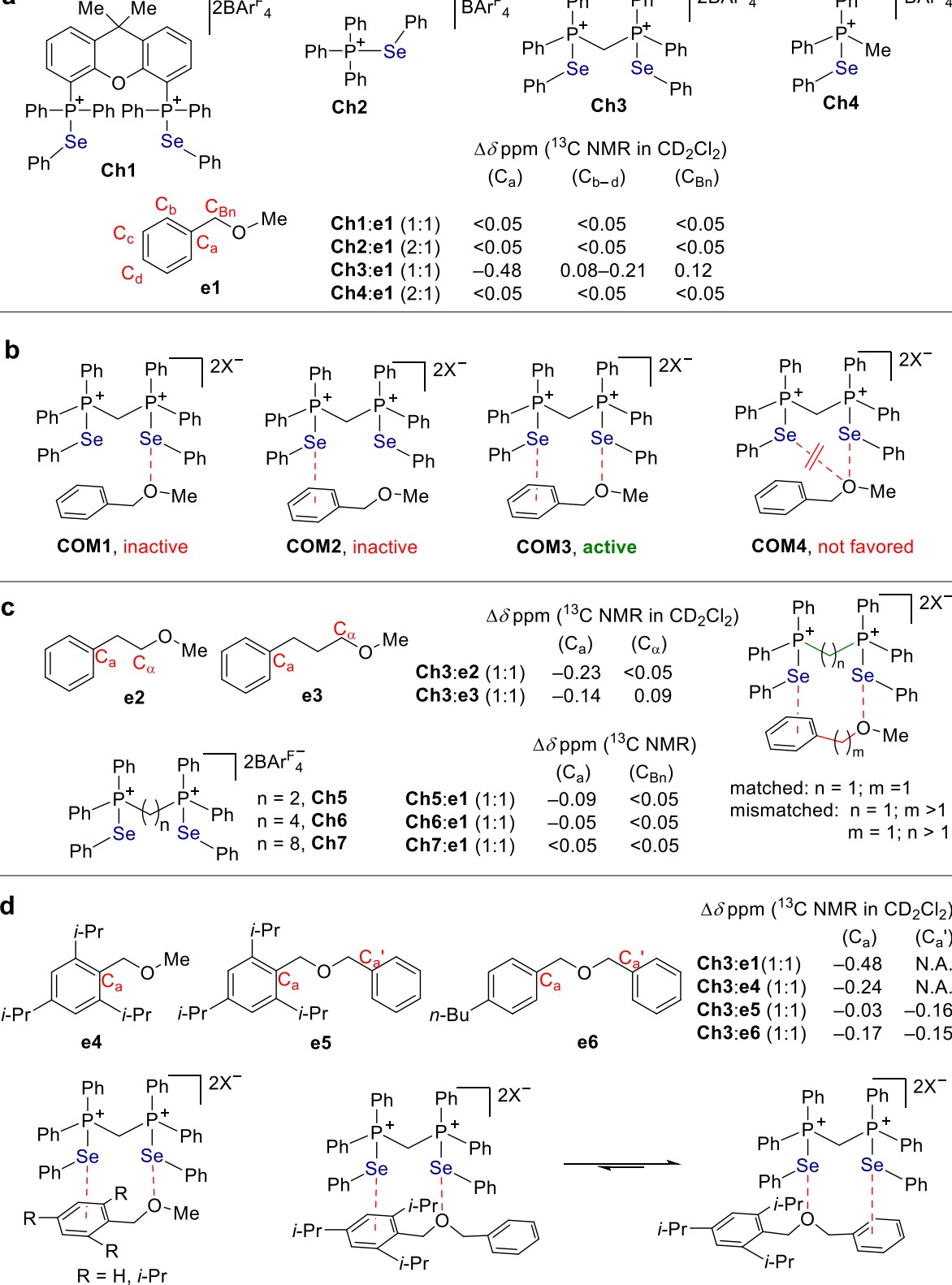

**Fig. 2 | Chalcogen bonding interaction with ethers (concentration: 0.10 M⁻¹ for bidentate donor; 0.10 M⁻¹ for ether; 0.20 M⁻¹ for monodentate donor). a** The interaction between chalcogen bonding donors and **e1**. **b** The interaction modes between chalcogen bonding donors and ethers. **c** The match and mismatch of binding sites between chalcogen bonding donors and ethers. **d** Molecular control experiments. BArᶠ₄⁻: tetrakis[3,5-bis (trifluoromethyl)phenyl]borate.

+0.21 ppm). Since $C_a$ is attached to the benzylic carbon, the above observation could be rationalized by electron–withdrawing via Se···π bonding which lowers the π* energy, thus leading to an increased σ→π* charge transfer. As a result, a marked shielding of $C_a$ was observed owing to an increased electron density of $C_a$, which in turn contributes to the cleavage of C–O bond.

To understand the chalcogen bonding interaction with ether, density functional theory (DFT) calculations were carried out at

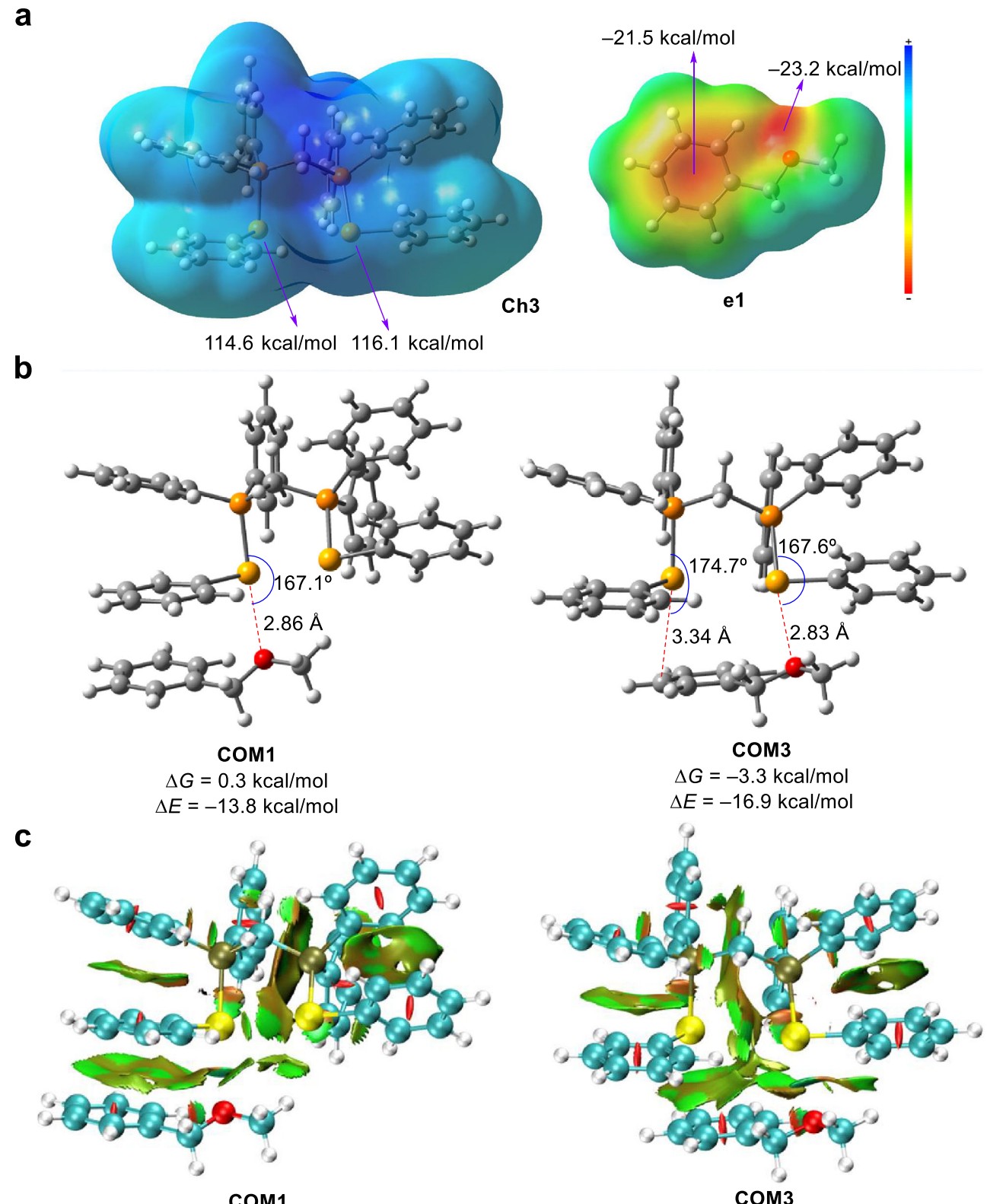

**Fig. 3 | DFT calculations. a** Electrostatic potential map. **b** Optimized complexes. **c** NCI analysis of complexes.

M06-2X/6-311 g(d,p) level of theory corrected with the Grimme's dispersion (D3)[44]. The electrostatic potential map in Fig. 3a suggests that the two selenium atoms have highly electropositive sites and the electrostatic potential values are +114.6 and +116.1 kcal/mol, respectively. As shown in Fig. 3b, in contrast to the Se···O bonding complex **COM1**, the results of DFT calculations suggest that an additional Se···π

interaction in **COM3** could significantly lower the energy barrier to generate the selenide-ether bonding complex. Further NCI analysis in Fig. 3c shows main noncovalent interactions. Light blue regions representing relatively strong interactions between Se···O exist in both the complexes of **COM1** and **COM3**, indicating that Se···O interactions contribute significantly to the formation of these supramolecules.

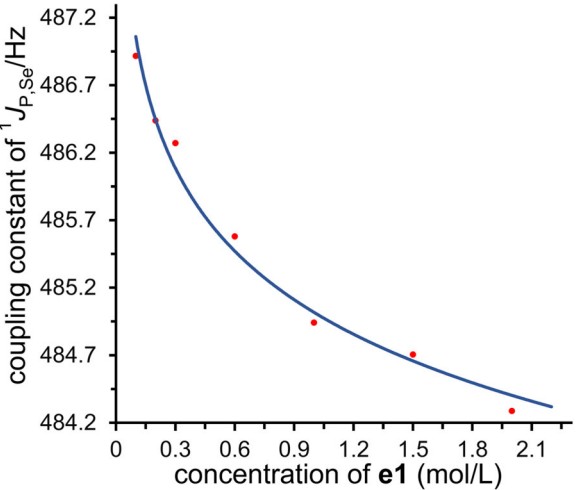

**Fig. 4 | The variation of coupling constant.** Relationship between the coupling constant of $^{1}J_{P,Se}$ of **Ch3** and the concentration of **e1**.

While π···π interaction could be observed in **COM1**, the computational results suggest it is more favarouble to generate **COM3** owing to the contribution of Se···π interaction.

In principle, the increasing of the concentration of ether **e1** results in the generation of a higher concentration of equilibrating supramolecular bonding complexes. The donation of electron from ether to selenium would change the electron distribution around selenium, thus potentially having an effect on the spin-spin coupling. A significant observation is that the concentration of ether **e1** has a correlation with the variation of coupling constant $^{1}J_{P,Se}$ (Fig. 4). A general trend is that a higher concentration of **e1** resulted in a more dramatic variation of the coupling constant. In the presence of 2.0 M **e1**, the coupling constant $^{1}J_{P,Se}$ varies as much as 2.2 Hz in contrast to **Ch3** itself.

To provide some information on the chalcogen bonding between selenide and ether, chalcogen bonding donor **Ch8** bearing a benzylic ether fragment was synthesized (Fig. 5). The X-ray crystal structure of **Ch8** shows the presence of an intramolecular Se···O interaction, wherein the distance between Se and O is 2.77 Å which is markedly shorter than the sum of their van der Waals radii 3.42 Å while the interaction angle of C-Se···O is 165.3°. For intermolecular interactions,

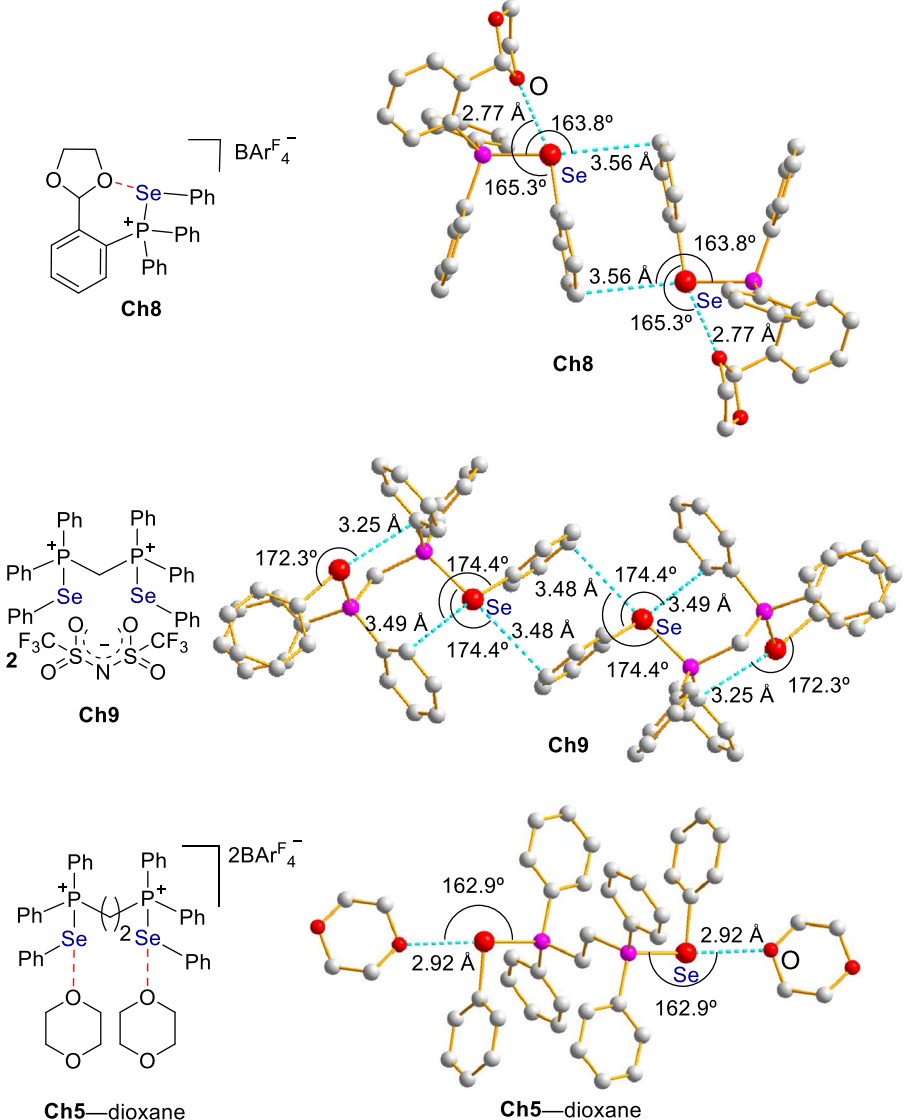

**Fig. 5 | Structure and bonding.** Intramolecular Se···O and intermolecular Se···π interactions in the crystal structure of **Ch8**; Intra- and intermolecular Se···π interactions in the crystal structure of **Ch9**; Intermolecular Se···O interactions in the cocrystal structure of **Ch5** and dioxane. Hydrogens and counteranions were omitted in these X-ray crystal structures for clarity.

| cat. | 3 (%); 2 h | 3 (%); 3 h | 3 (%); 5 h | 3 (%); 8 h | 3 (%); 12 h |
|------|-----------|-----------|-----------|-----------|------------|
| Ch1 | <5% | <5% | <5% | <5% | <5% |
| Ch2[a] | <5% | <5% | 7% | 13% | 15% |
| Ch3 | 18% | 32% | 62% | 74% | 87% |
| Ch4[a] | <5% | <5% | <5% | <5% | <5% |
| Ch5 | 10% | 26% | 44% | 51% | 58% |
| Ch6 | <5% | 11% | 21% | 33% | 46% |
| Ch7 | <5% | 7% | 16% | 23% | 25% |

[a]20 mol % of **Ch2** or **Ch4** was used.

**Fig. 6 | Structure and catalytic activity.** Distinct catalytic activity for chalcogenides **Ch1**-**7**. Catalytic performance for reference hydrogen- and halogen-bonding catalysts **H1**, **H2** and **XB**. DCE: 1, 2-dichloroethane.

Se···π bonding interactions are more favorable than Se···O interactions in the packing diagram of **Ch8**.

To further unveil the intermolecular Se···π bonding interaction, the less coordinating $BAr^F_4{}^-$ counteranion in **Ch3** was replaced by triflimide, i.e. $(CF_3SO_2)_2N^-$ (Fig. 5). Despite the presence of multiple oxygen binding sites in counteranion, the crystal structure of **Ch9** shows that the intermolecular interactions were dominated by Se···π bonding interactions. On the other hand, the co-crystal structure of **Ch5** and 1,4-dioxane shows the presence of a bonding complex operated through the intermolecular Se···O interactions (Fig. 5). These crystal structures depicted in Fig. 5 suggest it is possible for this type of selenides to interact with an ether to form Se···O and Se···π bonding interactions.

**Supramolecular catalysis with ethers**

For the activation of benzylic ethers, the observations in Fig. 2 suggest that (1) bidentate **Ch3** is more efficient than the other counterparts, and (2) monodentate **Ch2, 4** and bidentate **Ch1** with a rigid backbone, which are difficult to interact with both the oxygen atom and the phenyl ring in ether simultaneously, are not productive. Furthermore, bidentate selenides **Ch5-7** with a longer linker are less effective. To verify such a relationship between the structure and catalytic activity, the Friedel-Crafts reaction between ether **1a** and *m*-xylene **2a** was employed (Fig. 6)[13–18]. In the absence of a promoter, the reaction between **1a** and **2a** did not take place. In line with the above observations, catalyst **Ch1** was inactive in this reaction, while **Ch3** gave the best results among these catalysts. Accordingly, catalysts **Ch2/4** showed no or low catalytic activity (Fig. 6). The catalytic activity decreased gradually upon increasing the length of linker from one carbon to eight carbons (**Ch5-7**).

Tracing the reaction system by $^{31}$P NMR indicated that catalyst **Ch3** was stable under this reaction condition and only $^{31}$P signal corresponding to **Ch3** was observed (see Supplementary Fig. 11). For reference, the other types of weak interaction donors were

investigated (Fig. 6). The experimental results showed that the representative hydrogen bonding catalysts **H1-2** and halogen bonding catalyst **XB**, had no catalytic activity. Furthermore, this reaction did not work in the presence of 10 mol % TsOH or CF₃COOH even at 50 °C.

The strength of chalcogen bonding interactions follows the order of S < Se[45]. Accordingly, a decreased catalytic activity in principle would be expected upon replacement of the selenium atom in **Ch3** with sulfur (Fig. 7a). In line with this notion, the experimental results indicated that sulfur-based donor **Ch10** gave a lower catalytic activity in contrast to **Ch3**. On the other hand, since S is more electronegative than Se, phosphorus part in **Ch10** is therefore more positively charged than that in **Ch3**, which is consistent with the results of $^{31}$P NMR experiments (42.30 ppm for **Ch10** vs 30.27 ppm for **Ch3**). Taking the structure and catalytic performance into consideration, the comparison between **Ch10** and **Ch3** suggests that the catalytic activity arises from the selenium part.

Since chalcogen bonding is a type of highly directional interaction, the steric hindrance of the selenium part could prevent the electron donors to approach selenium, thus deactivating the catalyst to some extent. Then a more steric catalyst **Ch11** was synthesized. In agreement with the chalcogen bonding property, the installation of isopropyl groups in the phenyl ring attached to selenium of **Ch3** led to a decreased catalytic activity (Fig. 7a). To provide further information, monodentate catalysts **Ch12** and **Ch13** was synthesized, and they showed no catalytic activity (Fig. 7b). The distinct catalytic activity of bidentate catalyst **Ch3** and these monodentate catalysts underscores the importance of a dual interaction. Further investigation using catalyst **Ch14** suggests that the attachment of a phenyl group to selenium is not essential since a methyl group attached to selenium was catalytically active, and product **3** was obtained in 41% yield (Fig. 7b). We further tested one of the most active catalysts that we have ever obtained, i.e. catalyst **Ch15**[43]. Because **Ch15** could not interact with benzylic ether via a dual interaction approach, it showed no catalytic activity (Fig. 7b).

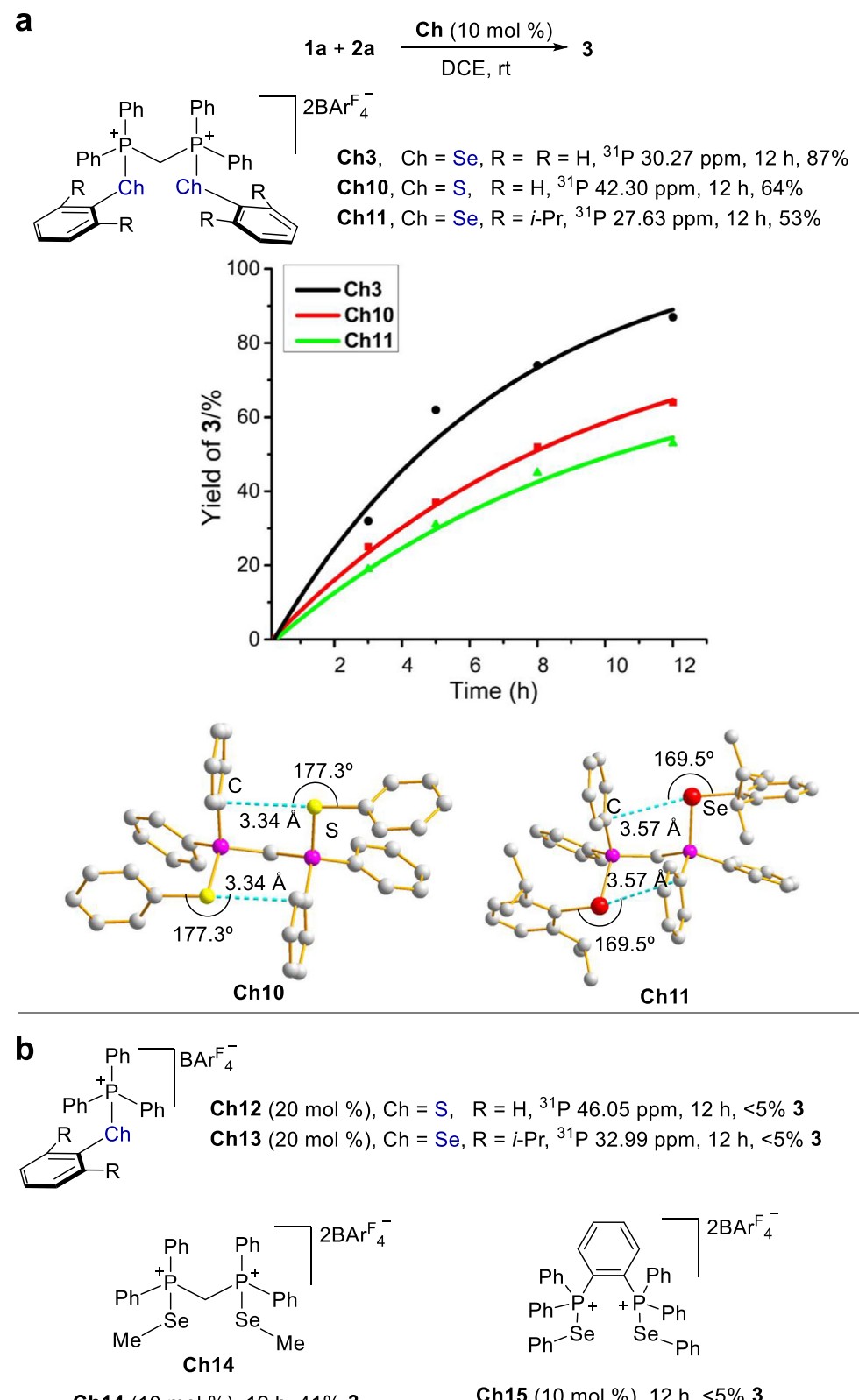

**Fig. 7 | Structure and catalytic activity. a** Comparing the catalytic activity of analogous catalysts **Ch3**, **Ch10**, and **Ch11**. **b** Catalytic activity of reference catalysts **Ch12-15**.

These additional results indicate the essential role of a dual activation approach.

The experimental results showed that a diverse array of alkoxide ions including MeO⁻, BnO⁻, PhO⁻ and *n*-PrO⁻ could serve as effective leaving groups (Fig. 8a). For substrate **1c**, the $^{13}$C NMR analysis of 1:1 mixture of **1c** and **Ch3** in CD$_2$Cl$_2$ revealed that the perturbation of the

$^{13}$C signals is similar upon comparing the two phenyl rings of **1c**. For this chalcogen bonding catalysis approach, the complexation of catalyst and ether is an essential process but is not the rate-determining step. Once the supramolecular bonding complex is formed, the more favorable C-O bond cleavage enabling the formation of a more stable carbocation determines the reaction outcome. Moreover, a range of

**Fig. 8 | Chalcogen bonding catalytic transformation of ethers. a** Alkoxide leaving groups. **b** Coupling reactions.

siloxides such as $Me_3SiO^-$, $t\text{-}BuMe_2SiO^-$ and $t\text{-}BuPh_2SiO^-$ were also demonstrated to be effective leaving groups. For substrate **1g**, phenol would be generated along with the desirable product **5**, which could in turn competitively react with ether **1g** to give a side product (*vide infra*, compound **19**). Further investigation revealed that both benzylic and allylic ethers could be activated to participate in different reaction processes, giving products **6-22** in moderate to good yields (Fig. 8b). Upon addition of 1.0 equiv $CaH_2$ to the catalysis system, product **6** was obtained in 76% yield and the reaction efficiency was not affected. Furthermore, this reaction did not work in the presence of 10 mol % TsOH or $CF_3COOH$. For products **8** and **19**, their ortho-selective regioisomers were obtained in 14% and 16% yields, and the isomeric ratios are 1:4.6 and 1:4.2, respectively.

Generally, alkoxide ions are considered as poor leaving groups to engage in elimination reactions. Instead of trapping with a nucleophile, the in situ generated carbocation could undergo an elimination process. This dual Se⋯O and Se⋯π bonding activation approach enables elimination reaction of ether to take place in a highly stereoselective manner. As shown in Fig. 9, a controlled synthesis of specific stereoisomer could be achieved using this method, and the elimination of

**Fig. 9 | Stereospecific elimination.** Chalcogen bonding catalysis approach to the elimination of ether **1u**. The use of $FeCl_3$ as a reference Lewis acid catalyst for the elimination of ether **1u**.

ether **1u** gave product (*E*)-**23** in 76% yield with 14.5:1 *E/Z* stereoselectivity. In contrast, a metal Lewis acid approach, i.e. $FeCl_3$, afforded lower selectivity (*E/Z* 8.2:1) in 79% yield.

Using Boc-protected pyrrolidine derivative **1v** as a reactant, an intramolecular cyclization reaction took place to give heterocycle **24** in 71% yield (Fig. 10a). Upon replacement of TMS with Me, substate **1w** underwent the same cyclization reaction to give product **24** in 73%

**Fig. 10 | Cyclization of ethers. a** Intramolecular cyclization reactions of **1v** and **1w**. **b** Intermolecular cyclization reaction of **1x**.

yield. Furthermore, under catalysis of **Ch3**, a cyclization reaction took place and ether **1x** was directly transformed to product **25** (Fig. 10b). Upon addition of either 1.0 equiv $K_2CO_3$ or $CaH_2$ to the catalysis system, the reaction proceeded with almost the same efficiency, and product **25** was obtained in 71% and 69% yields, respectively. Furthermore, tracing this reaction system by $^{31}P$ NMR revealed that catalyst **Ch3** remained unchanged since only $^{31}P$ signal corresponding to **Ch3** was observed (see Supplementary Fig. 13).

## Discussion

In conclusion, this work demonstrates that a rationally designed weak interaction mode is capable of activating relatively strong C-O σ-bonds of ethers, which is of vital importance for a sustainable development of supramolecular catalysis. The experimental results suggest a dual Se⋯π and Se⋯O bonding mode is in operation for the activation of benzylic as well as allylic ethers. This catalysis mode could be applied to a diverse array of transformations including coupling, elimination,

cyclization and the Friedel-Crafts reactions, thus enabling adding ethers in the list of supramolecular catalysis.

## Methods

### General procedure for chalcogen bonding catalytic transformation of ethers

To a reaction mixture of catalyst **Ch3** (10 mol %, 0.02 mmol) in a 10 mL-Schlenk tube was added DCE (1.0 mL) under argon atmosphere. Then ether substrate **1** (0.2 mmol) and nucleophile (0.6 mmol, 3.0 equiv) was added to the above reaction mixture. The reaction was stirred at room temperature until the completion of the reaction as judged by TLC analysis. Then the solvent was removed under reduced pressure, and the residue was purified by flash chromatography on silica gel to give the desired products.

## Data availability

The X-ray crystallographic data for the structures reported in this Article have been deposited at the Cambridge Crystallographic Data Centre, under deposition numbers 2203200 (**Ch5**-dioxane), 2203201 (**Ch8**), 2203203 (**Ch9**), 2203204 (**Ch10**), and 2203205 (**Ch11**). Copies of these data can be obtained free of charge via https://www.ccdc.cam.ac.uk/structures/. All the other data supporting the findings of this study are available within the article and its Supplementary Information file. Source data are provided with this paper.

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

## Acknowledgements

Y.W. acknowledges the financial support from the National Natural Science Foundation of China (NSFC, Grant Nos. 22022105, 21971147, 21772113), and the Natural Science Foundation of Shandong Province (ZR2019JQ08, ZR2020ZD35). Z.G.Z. acknowledges the financial support of the China Postdoctoral Science Foundation (2021TQ0192). P.P.Z. acknowledges the financial support of the Fundamental Research Funds for the Central Universities under Grant No. lzujbky-2021-sp43. We thank Prof. Di Sun at Shandong University for assistance with the X-ray crystal structure analysis. Technical support from SDU SC & PP research facilities was acknowledged.

## Author contributions

Y.W. conceived and directed the project and wrote the manuscript. Z.Z. (Zhiguo) conducted the experiments and prepared the Supplementary Materials. Z.Z. (Ziqiang) and Y.P. prepared part of substrates and catalysts and reproduced part of these catalysis reactions. P.P.Z. performed computational studies.

## Competing interests

The authors declare no competing interests.
