## [Peer Review File · Nature Communications]

Supramolecular Catalysis with Ethers Enabled by Dual Chalcogen Bonding ActivationReviewers' Comments:

Reviewer #1:

Remarks to the Author:

The manuscript "Supramolecular Catalysis with Ethers Enabled by Dual Chalcogen Bonding Activation" study the possibility of the activation of the ethers by weak, dual Se···n and Se···O interactions, and demonstrates its application in catalytic reactions. The design of new chalcogen bond catalysts is of great importance in organic synthesis and activation of very strong C-O bonds by the means of weak chalcogen interactions would be a significant contribution to this field of chemistry. The authors indeed presented evidence that their chalcogen-containing bidentate systems could be used as catalysts in this regard. The manuscript is well-written and logically structured, and experimental data seems well elaborated. However, I have some concerns regarding the analysis of noncovalent interactions in studied systems. Therefore, I suggest the publication of this manuscript after addressing the following comments:

1. All chalcogen-containing systems have chalcogen atoms attached to the Ph fragment, and all ethers also contain Ph fragments. In my opinion, this can result in π -stacking interactions between these fragments that could be significantly stronger than possible chalcogen interactions. A good example of this is the Ch8 structure (Figure 4). Authors claim that in this case, Se···n interactions are more favorable than Se···O interactions. In my opinion, Se···n interactions in this structure are coupled with π -stacking interactions between corresponding aromatic fragments. Moreover, it is questionable how significant is the contribution of Se···n interactions compared to mentioned π -stacking interactions. In neutral systems, π -stacking interactions would be significantly stronger. I believe that the most elegant way to overcome these problems would be to run a series of quantum chemical (or Density Functional Theory) calculations on selected model systems by which the energy and geometry of chalcogen bonds would be evaluated.

Also, the analysis of noncovalent bonding in crystal structures was predominantly based on the analysis of the sums of van der Waals radii. Although this is a useful indicator of possible noncovalent bonding, it can not be used as proof of the existence of noncovalent interactions in crystal structures, as recently pointed out by Politzer and Murray (Struct Chem 32, 623–629 (2021)). Along these lines (and as already suggested), quantum chemical or DFT calculations of energies of noncovalent interactions performed on suitable model systems would be very useful. Alternatively, even simple NCI (Noncovalent Interaction) index calculations and analysis of visualized NCI plots would probably be enough to support proposed noncovalent bonding patterns and complement them with additional information related to noncovalent bonding.

2. I also believe that the study would benefit from the calculations and analysis of the Molecular Electrostatic Potentials of studied molecules (although I do not insist on this). Assuming that these interactions are mostly driven by electrostatics, this type of analysis would be helpful for understanding the role of noncovalent contacts in studied systems. Moreover, visualization of σ -holes on chalcogen atoms could confirm proposed noncovalent bonding patterns.

Reviewer #2:

Remarks to the Author:

Wang reported a Friedel-Crafts reaction of benzyl or allyl ethers with arenes. Phosphonium selenide catalysts that are developed by the group were used in the study. The reaction scope covers a range of benzyl ethers (mainly disubstituted) and some allyl ethers. The referee judged the paper on the basis of applicability and conceptual novelty. However, the standard may not be up to the standard of Nat commun in the current form.

In terms of applicability, Friedel-Crafts reaction of benzyl ethers with arenes have already been widely

reported using different methodologies (many are not cited). The selectivity is comparable to that of the literature examples. It should be pointed out that benzylic C-O bonds are not that strong. Electron rich arenes are also used in this paper. So, both benzyl ethers and arenes are activated in this study.

As for the conceptual novelty, Wang proposed a dual activation mode in the paper but the referee has a major concern on this proposal. Friedel-Crafts reaction of benzyl ethers typically involve the generation of carbocation intermediate via the dissociation of the oxygen. So, it is reasonable that the catalyst coordinates to the oxygen. However, whether the Se- π interaction is responsible to the substrate activation is unclear.

It is stated in the paper that the sigma to π^* charge transfer is responsible to the shielding effect of Ca. This is too speculative. If sigma to π^* charge transfer is favored, it should also draw electron from the C-sp³ sigma bond as well, which should destabilize the benzylic carbocation intermediate in the Friedel-Crafts reaction. In addition, in figure 2A it is pointed out that the phenyl instead of the bulky iPr₃Ph coordinates to the Se due to less steric hindrance. If the dual activation mode is correct and the less hindered phenyl ring coordinates to the Se, the Friedel-Crafts reaction should proceed at the side with less hindered benzylic group. However, for substrate 1c the C-O bond cleaves as the bulkier benzylic carbon. One should note that the thermodynamically stable species observed in the NMR experiments does necessary to be the active species. All these data are pointing to the mechanism of Friedel-Crafts reactions that rely on the stabilization of carbocation intermediate with high substrate dependence.

Minor points: (1) Would the reaction be promoted by trace Bronsted acid? The reaction in Figure 9B is not relevant to those in Figure 7 and so the addition of acid scavenger to the reaction in figure 7 is needed to check this point. (2) For some relatively entries with lower yields, e.g. 1g, 8, 19, are there any regioisomers? (3) In figure 9A, instead of the formation of carbocation at the tertiary benzylic position, could the OTMS decompose to give OH, which could then react with the Boc carbonyl group to give product 24?

Reviewer #3:

Remarks to the Author:

In this manuscript, Wang et al have demonstrated an interesting work on developing a dual noncovalent bonding catalysis approach to achieve ether activation, which is a challenging goal in synthetic chemistry. The chalcogen catalyst showed rare observed ability of catalyzing the activation of inert C-O bond in ethers via unique synergistic Se \cdots n and Se \cdots O interactions, which enabled a range of transformation reactions. The catalysis mechanism was exhaustively evaluated by careful experimental design and the conclusions are basically convincing. In addition, the manuscript is well prepared and organized, but some experimental design and data presentation need to be improved. Overall, I think the discoveries in this work are important, and it could be considered for publication on Nature Communications after addressing the following issues.

1. The main novelty of this work is achieving the unique noncovalently catalyzed ether activation, to make a clearer demonstration, in the introduction part, I think the author should add suitable descriptions on the significance of ether activation in organic synthesis, summary of the current strategies, and particularly the peculiarity of the proposed dual Se \cdots n and Se \cdots O bonding catalysis approach.
2. The author should check the manuscript carefully to exclude confusing expressions, for example, last two sentences of paragraph 2 in page 3, different descriptive terms were used to describe the same chemical shift variation (<0.5 ppm). Furthermore, the tense was improperly used in these sentences.
3. For better presentation and readability, the matched and mismatched bonding models should be provided in Figures 2C and 2D.
4. The catalyst Ch3 was also applied in catalyzing elimination and cyclization reactions, and the

authors attributed these two applications to the dual Se···n and Se···O bonding catalysis, however, no direct evidences were provided to support this mechanism. Thus, additional experimental data and demonstrations are necessary in this part. And I also wondered to know how the products in figure 9 were formed in this catalysis reaction.

Point-by-point response to the reviewers' comments on NCOMMS-23-04292

Reviewer #1 (Remarks to the Author):

The manuscript "Supramolecular Catalysis with Ethers Enabled by Dual Chalcogen Bonding Activation" study the possibility of the activation of the ethers by weak, dual $\text{Se}\cdots\pi$ and $\text{Se}\cdots\text{O}$ interactions, and demonstrates its application in catalytic reactions. The design of new chalcogen bond catalysts is of great importance in organic synthesis and activation of very strong C-O bonds by the means of weak chalcogen interactions would be a significant contribution to this field of chemistry. The authors indeed presented evidence that their chalcogen-containing bidentate systems could be used as catalysts in this regard. The manuscript is well-written and logically structured, and experimental data seems well elaborated. However, I have some concerns regarding the analysis of noncovalent interactions in studied systems. Therefore, I suggest the publication of this manuscript after addressing the following comments:

Response: We appreciate this reviewer for all these instructive comments on our work. These comments are very helpful to improve the quality of this work.

1. All chalcogen-containing systems have chalcogen atoms attached to the Ph fragment, and all ethers also contain Ph fragments. In my opinion, this can result in π -stacking interactions between these fragments that could be significantly stronger than possible chalcogen interactions. A good example of this is the Ch8 structure (Figure 4). Authors claim that in this case, $\text{Se}\cdots\pi$ interactions are more favorable than $\text{Se}\cdots\text{O}$ interactions. In my opinion, $\text{Se}\cdots\pi$ interactions in this structure are coupled with π -stacking interactions between corresponding aromatic fragments. Moreover, it is questionable how significant is the contribution of $\text{Se}\cdots\pi$ interactions compared to mentioned π -stacking interactions. In neutral systems, π -stacking interactions would be significantly stronger.

I believe that the most elegant way to overcome these problems would be to run a series of quantum chemical (or Density Functional Theory) calculations on selected model systems by which the energy and geometry of chalcogen bonds would be evaluated.

Response: According to your suggestion, DFT calculations were carried out, and the results corroborate a dual $\text{Se}\cdots\pi$ and $\text{Se}\cdots\text{O}$ interaction mode (Figure 3). As shown in Figure 3B, in contrast to the $\text{Se}\cdots\text{O}$ and $\pi\cdots\pi$ bonding complex **COM1**, the results of DFT calculations suggest that the formation of $\text{Se}\cdots\text{O}$ and $\text{Se}\cdots\pi$ bonding complex **COM3** is more favorable and it has a significantly lower energy barrier to generate **COM3** owing to the contribution from $\text{Se}\cdots\pi$ interaction.

In addition to DFT calculations, to unambiguously exclude the role of $\pi\cdots\pi$ interaction between catalyst and substrate, we also carried out further experiments. It would provide significant insights into this point upon replacing the phenyl group attached to selenium by a methyl group. To this end, a new catalyst **Ch14** was synthesized. Although methyl is an electron-donating group in contrast to phenyl, the catalysis results indicate **Ch14** was catalytically active and product **3** was obtained in 41% yield. These new experimental results suggest $\pi\cdots\pi$ interaction is not essential. The new results were added in Figure 7.

We also made corresponding comments in the revised manuscript, please see the text with a yellow background: "Further investigation using catalyst **Ch14** suggests that the attachment of a phenyl group to selenium is not essential since a methyl group attached to selenium was catalytically active, and product **3** was obtained in 41% yield."

Also, the analysis of noncovalent bonding in crystal structures was predominantly based on the analysis of the sums of van der Waals radii. Although this is a useful indicator of possible noncovalent bonding, it can not be used as proof of the existence of noncovalent interactions in crystal structures, as recently pointed out by Politzer and Murray (Struct Chem 32, 623–629 (2021)). Along these lines (and as already suggested), quantum chemical or DFT calculations of energies of noncovalent interactions performed on suitable model systems would be very useful. Alternatively, even simple NCI (Noncovalent Interaction) index calculations and analysis of visualized NCI plots would probably be enough to support proposed noncovalent bonding patterns and complement them with additional information related to noncovalent bonding.

Response: The interaction mode was demonstrated by DFT calculations of energies. In addition, NCI analysis was carried out to show main noncovalent interactions in the optimized complexes. These computational results corroborate a Se \cdots O and Se \cdots π bonding mode, please see Figures 3B and 3C.

2. I also believe that the study would benefit from the calculations and analysis of the Molecular Electrostatic Potentials of studied molecules (although I do not insist on this). Assuming that these interactions are mostly driven by electrostatics, this type of analysis would be helpful for understanding the role of noncovalent contacts in studied systems. Moreover, visualization of σ -holes on chalcogen atoms could confirm proposed noncovalent bonding patterns.

Response: Molecular Electrostatic Potentials were calculated. The electrostatic potential map in Figure 3A suggests that the two selenium atoms have highly electropositive sites (σ -holes) and the electrostatic potential values are +114.6 and +116.1 kcal/mol, respectively.

Reviewer #2 (Remarks to the Author):

Wang reported a Friedel-Crafts reaction of benzyl or allyl ethers with arenes. Phosphonium selenide catalysts that are developed by the group were used in the study. The reaction scope covers a range of benzyl ethers (mainly disubstituted) and some allyl ethers. The referee judged the paper on the basis of applicability and conceptual novelty. However, the standard may not be up to the standard of Nat commun in the current form.

Response: We appreciate this reviewer for these instructive comments on our work. Based on these comments, we designed and carried out a series of new experiments, which are very helpful to improve the quality of this work.

In terms of applicability, Friedel-Crafts reaction of benzyl ethers with arenes have already been widely reported using different methodologies (many are not cited). The selectivity is comparable to that of the literature examples. It should be pointed out that benzylic C-O bonds are not that strong. Electron rich arenes are also used in this paper. So, both benzyl ethers and arenes are activated in this study.

Response: With regard to the topic on the activation of benzylic ether C-O σ -bonds by weak interactions, we would like to elucidate the following points.

(1) *It remains an unresolved topic in supramolecular catalysis.* Catalysis with weak interactions largely relies on the activation of comparatively reactive π -bonded functional groups, with carbonyl, imine, and nitro groups as dominant targets, while the activation of relatively strong benzylic or allylic ether C-O σ -bond by weak interactions remains an unresolved topic. In this context, the activation of relatively strong σ -bond by weak interactions has been recognized as a fundamentally unresolved challenge in supramolecular catalysis.

(2) *The activation of benzylic C-O σ -bond represents a long-standing challenge for weak interactions.* For known weak interactions, hydrogen bonding catalysis has been extensively studied for over 20 years while halogen bonding catalysis has been investigated for more than 10 years, however, there is yet no example on the activation of C-O σ -bond using these well-appreciated noncovalent forces, regardless of benzylic or allylic ethers. Therefore, it is not surprised that when we used representative hydrogen bonding and halogen bonding donors to activate benzylic ethers, none of these reference catalysts is active (Figure 5). In contrast to the established activation targets such as carbonyl π -bond, C-O σ -bond is comparatively strong while alkoxide is a poor leaving group, thus setting up a hard-to-reach barrier for weak interactions to activate ethers. Even though few reports indicate that strong metal Lewis acids and superacid could activate benzylic ether C-O σ -bonds, however, upon taking superacid $\text{BF}_3\text{-H}_2\text{O}$ as a metal-free reference for comparison (please see ref 13, *Green Chem.* **2014**, *16*, 2976–2981), the coupling reactions between benzylic ethers and electron rich arenes requires a very harsh reaction condition—that is—the use of stoichiometric amount superacid $\text{BF}_3\text{-H}_2\text{O}$ (1.2 equiv) and high temperature (120 °C). These relevant literatures have been commented and added in the revised manuscript. These examples imply that the coupling reactions between benzylic ethers and electron rich arenes is not trivial even for superacid. Notably, this chemistry shows wide potential of this chalcogen bonding catalysis approach, which was not limited to the reactions between benzylic ethers and arenes but it can be applied to many reaction types such as transesterification, elimination and cyclization reactions.

(3) *Chalcogen bonding catalysis approach to the activation of ethers has its unique advantage in the controlling of reaction selectivity.* Chalcogen bonding donors and strong metal Lewis acids belong to different disciplines and they have distinct chemical properties. Conventionally, divalent chalcogenides are used as Lewis bases which react with electrophiles to form covalent bonds. Notably, chalcogen bonding catalysis is still in its infancy, and the capability of this new catalysis concept yet remains to be explored. In the revised manuscript, the advantage of chalcogen bonding catalysis approach was

demonstrated by its capability in the controlling of the stereoselectivity. In contrast to the strong Lewis acid approach, the chalcogen bonding approach catalysis approach can give significantly enhanced stereoselectivity in the elimination of ether, please see Figure 9. These new results highlight the potential of chalcogen bonding catalysis, which were added and commented in the revised manuscript.

As for the conceptual novelty, Wang proposed a dual activation mode in the paper but the referee has a major concern on this proposal. Friedel-Crafts reaction of benzyl ethers typically involve the generation of carbocation intermediate via the dissociation of the oxygen. So, it is reasonable that the catalyst coordinates to the oxygen. However, whether the Se- π interaction is responsible to the substrate activation is unclear. It is stated in the paper that the σ to π^* charge transfer is responsible to the shielding effect of Ca. This is too speculative. If σ to π^* charge transfer is favored, it should also draw electron from the C-sp³ σ bond as well, which should destabilize the benzylic carbocation intermediate in the Friedel-Crafts reaction.

Response: Thank you for your comment. Mechanistically, this question relates to the following two steps: (1) before C-O cleavage; (2) after C-O cleavage.

(1) Before C-O cleavage. For a benzylic or allylic system, hyperconjugation is a classic theory to describe such molecules, and there is σ - π hyperconjugation between benzylic or allylic σ -bonds and π system. Herein, hyperconjugation is the delocalization of electron-density that occurs when σ -bonded electrons contribute electron density to a π system. As a result, such σ - π hyperconjugation weakens the strength of benzylic or allylic σ -bonds, thus making these σ -bonds more reactive in contrast to their fully saturated counterparts. In this work, the imposing of a transient Se- π interaction would lower the π^* energy, which enhances the σ - π interaction and facilitates the σ - π electron density delocalization (including $\sigma_{(C-O)}-\pi$), thus further weakening the C-O σ -bond. The NMR observations support this transient Se- π interaction assisted electron density delocalization. This σ - π electron-density delocalization results in a more electron-rich aromatic carbon C_a (neighboring carbon of the benzylic carbon), which in turn stabilizes the benzylic carbon. Therefore, both the Se-O and Se- π interactions enable weakening the C-O bond, thus contributing to the cleavage of C-O bond.

(2) After C-O cleavage. Once the C-O bond is cleaved, the selenide catalyst would not interact with the electron-deficient benzylic carbocation anymore. It is a "free" carbocation. This is reasonable because in this case it is the repulsion other than attraction between these two electron-deficient species. On the other hand, except the in situ generated trace amount of transient carbocation intermediate, a much higher concentration of stable electron-rich ether starting material still exists in the reaction system. Upon considering the competitive bonding between these two species with catalyst, it is apparently that the interaction between ether and catalyst would suppress the interaction between catalyst and carbocation, no matter from the viewpoint of the concentration or electronic property of ether and carbocation.

Moreover, once the carbocation is generated, it is p- π conjugation that stabilizes the carbocation, and there is no σ - π conjugation anymore since the carbocation is a planar

structure. Therefore, the Se- π interaction is only a transient interaction to induce the C-O cleavage, and there is no Se- π interaction anymore after C-O cleavage. It is necessary to distinguish such a transient Se- π interaction with a permanent electron-withdrawing substitution group which is installed into a π system. It is unfavorable for the latter π system to generate a carbocation.

In addition, in figure 2A it is pointed out that the phenyl instead of the bulky iPr3Ph coordinates to the Se due to less steric hindrance. If the dual activation mode is correct and the less hindered phenyl ring coordinates to the Se, the Friedel-Crafts reaction should proceed at the side with less hindered benzylic group. However, for substrate **1c** the C-O bond cleaves as the bulkier benzylic carbon. One should note that the thermodynamically stable species observed in the NMR experiments does necessary to be the active species. All these data are pointing to the mechanism of Friedel-Crafts reactions that rely on the stabilization of carbocation intermediate with high substrate dependence.

Response: For the activation of substrate **1c**, it is also consistent with a dual Se-O and Se- π interaction mode. It should be noted that there is no substituent on the two phenyl groups in **1c**, therefore, it is anticipated that the selenide catalyst would interact with both phenyl group with nearly equal opportunity. To demonstrate this, we performed NMR experiments using 1:1 **1c** and **Ch3** in CD₂Cl₂. As expected, upon comparing the ¹³C signals of the corresponding carbons on the two phenyl rings, it was found that the perturbation of the ¹³C signals is almost the same for the two phenyl rings. We commented in the manuscript that: “¹³C NMR analysis of 1:1 mixture of **1c** and **Ch3** in CD₂Cl₂ revealed that the perturbation of the ¹³C signals is similar upon comparing the two phenyl rings of **1c**.” Therefore, for the interaction with either side of ether **1c**, the Se-O interaction is the same while the Se- π interaction is similar, the formation of a relatively stable secondary carbocation is more favorable in this case.

The main theme of this work—is that—the cleavage of relatively strong C-O σ -bond by weak interactions, particularly in the context that the other weak interactions are failed to execute such a duty, which is important and significant to the sustainable development of supramolecular catalysis. After the C-O σ -bond is cleaved, carbocation could react with various types of reactants through different reaction pathways. Notably, this work is just a starting point. It is anticipated that the activation of ethers by chalcogen bonding interactions would open up a new window for supramolecular catalysis, upon considering the fact that numerous reactions could be established on each functional group (carbonyl, imine, and nitro etc.) in supramolecular catalysis. Furthermore, we hope the demonstration that the use of emerging chalcogen bonding can activate relatively strong C-O σ -bond would inspire the development of more and more challenging transformations on this topic.

Minor points: (1) Would the reaction be promoted by trace Bronsted acid? The reaction in Figure 9B is not relevant to those in Figure 7 and so the addition of acid scavenge to the reaction in figure 7 is needed to check this point.

Response: We tried two representative reactions in Figures 6 and 8 using TsOH or TFA as

a catalyst, and no reaction took place. We have commented in the manuscript that: "Furthermore, this reaction did not work in the presence of 10 mol % TsOH or CF₃COOH even at 50 °C." Upon addition of stoichiometric amount of CaH₂ into the coupling reaction as shown in Figure 7 (renamed as Figure 8 in the revised manuscript), the reaction worked with almost the same efficiency. We made a comment in the revised manuscript that: "Upon addition of 1.0 equiv CaH₂ to the catalysis system, product **6** was obtained in 76% yield and the reaction efficiency was not affected."

(2) For some relatively entries with lower yields, e.g. **1g**, **8**, **19**, are there any regioisomers?

Response: For substrate **1g**, the leaving group is phenol. We found that side reactions that involve phenol as a competitive coupling partner took place, which affects the yield of desirable product. For **8** and **19**, there are regioisomers. We commented in the revised manuscript that: "For products **8** and **19**, their ortho-selective regioisomers were obtained in 14% and 16% yields, respectively."

(3) In figure 9A, instead of the formation of carbocation at the tertiary benzylic position, could the OTMS decompose to give OH, which could then react with the Boc carbonyl group to give product **24**?

Response: In the reaction system, the generation of OH by decomposing of OTMS was not observed. To unambiguously investigate this case, we prepared an inert OMe ether, i.e. **1w** in Figure 10A, and the cyclization reaction also occurred to give cyclization product **24** in 73% yield, which excludes a free OH approach. The new results were added in the revised Figure 10A and commented in the manuscript.

Reviewer #3 (Remarks to the Author):

In this manuscript, Wang et al have demonstrated an interesting work on developing a dual noncovalent bonding catalysis approach to achieve ether activation, which is a challenging goal in synthetic chemistry. The chalcogen catalyst showed rare observed ability of catalyzing the activation of inert C-O bond in ethers via unique synergistic Se···π and Se···O interactions, which enabled a range of transformation reactions. The catalysis mechanism was exhaustively evaluated by careful experimental design and the conclusions are basically convincing. In addition, the manuscript is well prepared and organized, but some experimental design and data presentation need to be improved. Overall, I think the discoveries in this work are important, and it could be considered for publication on Nature Communications after addressing the following issues.

Response: We appreciate this reviewer for all these instructive comments on our work. These comments are very helpful to improve the quality of this work.

1. The main novelty of this work is achieving the unique noncovalently catalyzed ether activation, to make a clearer demonstration, in the introduction part, I think the author should add suitable descriptions on the significance of ether activation in organic synthesis,

summary of the current strategies, and particularly the peculiarity of the proposed dual $\text{Se}\cdots\pi$ and $\text{Se}\cdots\text{O}$ bonding catalysis approach.

Response: We made a more detailed description on the ether activation, please see: "As a representative case, even though benzylic as well as allylic ethers are frequently used reactants in organic synthesis, the activation of these ethers is not trivial. Generally, metal catalysts, superacid or strong photochemical conditions were employed to activate benzylic and allylic ethers.¹³⁻¹⁹ Owing to the relatively strong C-O σ -bonds and the intractable alkoxide leaving group,²⁰ it appears rather difficult for a metal-free approach to the activation of ethers, for instance, the coupling reactions between benzylic ethers and electron rich arenes requires the use of a stoichiometric amount of superacid $\text{BF}_3\text{-H}_2\text{O}$ and a high temperature (120 °C).¹³ In this context, the practices in supramolecular catalysis suggest the activation of ethers by weak interactions is an unfeasible approach,¹⁻¹² and it remains an unresolved topic (Figure 1A). Apparently, the comparatively strong C-O σ -bond sets up a hard-to-reach barrier for weak interactions to play an effective role in activating ethers.¹²"

As for the advantage of dual $\text{Se}\cdots\pi$ and $\text{Se}\cdots\text{O}$ bonding catalysis approach, we commented in the revised manuscript that: "Considering the substantial difference between strong covalent activation and weak interaction approaches, the dual chalcogen bonding catalysis approach has its unique advantage, which proceeds in the absence of any metal additive while it can give reactivity and control the reaction selectivity."

2. The author should check the manuscript carefully to exclude confusing expressions, for example, last two sentences of paragraph 2 in page 3, different descriptive terms were used to describe the same chemical shift variation (<0.5 ppm). Furthermore, the tense was improperly used in these sentences.

Response: We carefully checked the whole manuscript, and the confusing/improper descriptions were modified.

3. For better presentation and readability, the matched and mismatched bonding models should be provided in Figures 2C and 2D.

Response: The matched and mismatched bonding models were added in the revised Figures 2C and 2D.

4. The catalyst Ch3 was also applied in catalyzing elimination and cyclization reactions, and the authors attributed these two applications to the dual $\text{Se}\cdots\pi$ and $\text{Se}\cdots\text{O}$ bonding catalysis, however, no direct evidences were provided to support this mechanism. Thus, additional experimental data and demonstrations are necessary in this part.

Response: The elimination and cyclization reactions proceeded through the similar activation mode, since these substrates are also benzylic ethers. That is, the dual $\text{Se}\cdots\pi$ and $\text{Se}\cdots\text{O}$ bonding interaction between these ethers and catalyst lead to the C-O bond

cleavage, thus generating carbocations. The difference lies in the following transformations after C-O bond cleavage. Once the carbocations were generated, many different reaction pathways could take place, including coupling, elimination, and cyclization reactions. We have made more description in these parts. For example, we commented in the elimination part that: "Instead of trapping with a nucleophile, the in situ generated carbocation could undergo an elimination process. This dual $\text{Se}\cdots\text{O}$ and $\text{Se}\cdots\pi$ bonding activation approach enables elimination reaction of ether to take place in a highly stereoselective manner." An activation mode was given in Figure 10A, and Figure 10B follows the similar activation approach.

And I also wondered to know how the products in figure 9 were formed in this catalysis reaction.

Response: The mechanism for the generation of products **24** and **25** were given in the revised Figure 9 (renamed as Figure 10).

Reviewers' Comments:

Reviewer #1:

Remarks to the Author:

I believe that the paper is now suitable for publication in its current form. The authors have addressed all of my comments and conducted additional calculations and experiments, clarifying any potential ambiguities.

Reviewer #2:

Remarks to the Author:

In this revised paper, the authors addressed some of the questions raised by the referee. There is still a main concern by the referee about the proof of the dual activation mode, which is the key point in the paper. However, evidence on the dual activation mode is very weak.

It is suggested to try e5 or analogues that have phenyl substituents with steric bias in the reaction. Since it is believed that e5 could form complex with Ch3 preferentially at the less hindered benzyl ether, the C-O bond at the less hindered side should be activated and one would expect that Friedel-Crafts reaction should happen at the unsubstituted benzyl ether unit if the "Se-pi interaction" is the key factor. Similarly, for substrate 1c, it is proposed that it can form complex with Ch3 and the two phenyl rings in 1c have almost equal interaction with Ch3. As a result, one would expect that 1c should give a 1:1 product mixture in the Friedel-Crafts reaction. However, product 3 was obtained as the major product that is a typical Friedel-Crafts reaction product. Could COM1 in figure 2B be feasible? It is stated that COM1 can be ruled out because Ch2 is not interacting with e1. However, one should note that Ch3 can form a mono-dentate complex with oxygen using one Se while another Se-phosphonium cation is just a strong electron withdrawing group that enhances the first chalcogen bond.

Other points:

1. The reaction is obviously a Friedel-Crafts reaction. However, this named reaction is not indicated in the manuscript's title and nowhere in the paper.
2. The NMR study on a mixture of 1c and Ch3 is not indicated in SI.
3. For 1g, it is explained that phenol is competing with compound 4 and side product was obtained. What's the structure of the side product? Is it compound 19? The side product structure should be indicated in figure 8A.
4. For the regio-isomers in 8 and 19, the isomeric ratios should be indicated in figure 8B.

Reviewer #3:

Remarks to the Author:

In this revised manuscript, the authors have satisfactorily responded my questions raised in the previous version, and I suggest the acceptance for publication after addressing the other reviewers' concerns.

Point-by-point response to the reviewers' comments on NCOMMS-23-04292A

Reviewer #1 (Remarks to the Author):

I believe that the paper is now suitable for publication in its current form. The authors have addressed all of my comments and conducted additional calculations and experiments, clarifying any potential ambiguities.

Response: We appreciate this reviewer for the positive comments on our work.

Reviewer #2 (Remarks to the Author):

In this revised paper, the authors addressed some of the questions raised by the referee. There is still a main concern by the referee about the proof of the dual activation mode, which is the key point in the paper. However, evidence on the dual activation mode is very weak.

Response: We appreciate this reviewer for further comments on our work.

To strength the importance of a dual activation, in the revised manuscript, we further tested one of the most active catalysts that we have ever obtained (see *Angew. Chem. Int. Ed.* **61**, e202203671), i.e. catalyst **Ch15** in Figure 7. Because this catalyst could not interact with benzylic ether via a dual interaction approach, it showed no catalytic activity. These additional results highlight the essential role of a dual activation approach. The new results were added and commented in the manuscript.

The dual activation mode was supported by the following points:

- 1) The marked difference of catalytic activity between monodentate and bidentate catalysts. The two-fold amount of monodentate catalyst showed no or very low catalytic activity while their bidentate counterparts showed high catalytic activity, which indicates the importance of a dual activation approach.
- 2) As for bidentate catalyst, the structure has a decisive role. While catalysts **Ch1** or **Ch15** was efficient in different reactions, however, it was not capable of activating ethers in this case. The reason could be attributed to that the structure does not allow a simultaneous dual interaction with both phenyl and oxygen. In addition, catalysts **Ch3**, **Ch5-7** showed gradually decreased catalytic activity, because the increased distance between two selenium atoms has a negative effect on a dual interaction approach. The catalytic performance of these catalysts implies that a dual activation approach is operative.
- 3) A range of molecular control experiments suggest that the formation of the supramolecular bonding complex between catalyst and substrate via a dual interaction mode is facile. In addition, the DFT calculations corroborated a dual interaction.
- 4) The imposing of a transient Se- π interaction would lower the π^* energy, which enhances the σ - π interaction (including $\sigma_{(C-O)-\pi}$), thus further weakening the C-O σ -bond. It should be noted that the most marked variation of ^{13}C NMR not only comes

from the aromatic carbon C_a (neighboring carbon of the benzylic carbon), but more significantly this carbon shifts up-field as much as -0.48 ppm. The σ - π electron-density delocalization results in a more electron-rich aromatic carbon C_a , which in turn stabilizes the benzylic carbon. Therefore, a dual activation mode contributes to the cleavage of C-O bond.

In supramolecular catalysis, the identification of active weak bonding supramolecular complex is very difficult. This challenge has been well-recognized by the scientific community, see (Chem. Soc. Rev., 2021, 50, 7681-7724): “when compared to the other topics of supramolecular science, it certainly poses greater difficulties and hurdles associated to the impossibility to know the precise structure of the transition states to be stabilized by interaction with supramolecular catalysts”.

In our paper, while a precisely active supramolecular complex was difficult to identify, we tried our efforts to recognize a reasonable supramolecular bonding complex based on the experimental and computational results, while the other possible bonding modes could not explain the experimental results. What’s important, the main purpose of this paper — is to demonstrate a fact that weak interactions can cleavage C-O σ -bond, a topic not been addressed in the past several decades, and this goal was achieved.

It is suggested to try **e5** or analogues that have phenyl substituents with steric bias in the reaction. Since it is believed that **e5** could form complex with **Ch3** preferentially at the less hindered benzyl ether, the C-O bond at the less hindered side should be activated and one would expect that Friedel-Crafts reaction should happen at the unsubstituted benzyl ether unit if the “Se- π interaction” is the key factor. Similarly, for substrate **1c**, it is proposed that it can form complex with **Ch3** and the two phenyl rings in **1c** have almost equal interaction with **Ch3**. As a result, one would expect that **1c** should give a 1:1 product mixture in the Friedel-Crafts reaction. However, product **3** was obtained as the major product that is a typical Friedel-Crafts reaction product.

Response: For a catalysis reaction, the complexation of catalyst and substrate is just a necessary process but often is not the rate-determining step. The subsequent bond formation or cleavage event frequently determines the overall reaction process. This is also true in our work; the transformation of ether mainly includes the following steps: 1) the interaction of catalyst and ether; 2) the cleavage of C-O σ -bond; and 3) the subsequent reactions. It should be noted that the weak interaction of catalyst and ether is just a preliminary step, and the formation of the supramolecular bonding complex does not mean that the reaction would occur. Meanwhile, in case that there is a competitive reaction regardless of intramolecular or intermolecular version, a major supramolecular complex does not mean that it would necessarily lead to a major product. For an S_N1 -type reaction, the reaction outcome is determined by the subsequent cleavage of inert C-O σ -bond.

Taking the interaction of catalyst **Ch3** and **e1** as an example, the formation of supramolecular complex **COM3** has a relative Gibbs free energy of -3.3 kcal/mol. Catalyst **Ch3** is capable of interacting with **e1** or more steric **e4** as shown in Figure 2, albeit in a different extent. Therefore, when an unsymmetrical ether such as **e5** or **1c** was involved, it would form an equilibrating system consisting of different bonding complexes between

catalyst and either head of **e5** or **1c**, with less steric head as the more favorable bonding site. Once the bonding complex is formed, the subsequent cleavage of C–O σ -bond dominates the reaction. For an intramolecular competition reaction of **e5** or **1c**, there is no direct correlation between the more ratio of a specific supramolecular complex and the more portion of the corresponding product, but instead the more favorable C–O bond cleavage determines the reaction outcome. The experimental results showed that the formation of more stable carbocation from ethers **e5** and **1c** dominates the reaction outcome. The reaction outcome of **e5** was added in the Supplementary Information, see pages S18, S110-111.

Could COM1 in figure 2B be feasible? It is stated that COM1 can be ruled out because Ch2 is not interacting with e1. However, one should note that Ch3 can form a mono-dentate complex with oxygen using one Se while another Se-phosphonium cation is just a strong electron withdrawing group that enhances the first chalcogen bond.

Response: In the revised manuscript, as shown in Figure 7, we further investigated catalyst **Ch15**, one of the most active catalysts we have ever obtained. Because this catalyst could not interact with ether via a dual interaction approach, it interacts with ether via a **COM1**-type mode while the rest Se-phosphonium cation serves as an electron-withdrawing group. The experimental result showed that it has no catalytic activity. Considering the catalysis results of all monodentate and bidentate catalysts, **COM1** is a plausible but less efficient activation mode.

Other points:

1. The reaction is obviously a Friedel-Crafts reaction. However, this named reaction is not indicated in the manuscript's title and nowhere in the paper.

Response: We have indicated the Friedel-Crafts-type reaction in the manuscript, see pages 3, 11, and 18. Our current work aims to establish a weak interaction-enabled C–O σ -bond activation approach. After the C–O σ -bond is cleaved, carbocation could react with various types of reactants through numerous reaction pathways including Friedel-Crafts, coupling, elimination and cyclization reactions, etc., which is not the focal point of this work.

2. The NMR study on a mixture of **1c** and **Ch3** is not indicated in SI.

Response: The NMR was added in SI, see Figure S10 and Table S12 in page S31.

3. For **1g**, it is explained that phenol is competing with compound **4** and side product was obtained. What's the structure of the side product? Is it compound **19**? The side product structure should be indicated in figure 8A.

Response: The side product is **19**. It was commented in the manuscript and the structure was indicated in Figure 8A.

4. For the regio-isomers in **8** and **19**, the isomeric ratios should be indicated in figure 8B.

Response: The isomeric ratios were added in figure 8B.

Reviewer #3 (Remarks to the Author):

In this revised manuscript, the authors have satisfactorily responded my questions raised in the previous version, and I suggest the acceptance for publication after addressing the other reviewers' concerns.

Response: We appreciate this reviewer for the positive comments on our work.

Reviewers' Comments:

Reviewer #2:

Remarks to the Author:

In this revision, most of the concerns by the referee are addressed. There is one suggestion on modifying the discussion section to make the argument more self-consistent. There is a long discussion in the rebuttal letter but many of the points do not appear in the manuscript.

In the manuscript, there is a major discussion on the correlation of chemical shift of Ca and the proposal of dual interaction and the complex is crucial for the reaction. However, in the rebuttal letter it is stated that "the formation of the supramolecular bonding complex does not mean that the reaction would occur" and "a major supramolecular complex does not mean that it would necessarily lead to a major product". Instead of just explaining in the rebuttal letter, it is suggested to refine the discussion sections by adding the points such as "the complexation of catalyst and substrate is just a necessary process but often is not the rate-determining step", "more favorable C-O bond cleavage determines the reaction outcome.", etc.

Point-by-point response to the reviewers' comments on NCOMMS-23-04292B

Reviewer #2 (Remarks to the Author):

In this revision, most of the concerns by the referee are addressed. There is one suggestion on modifying the discussion section to make the argument more self-consistent. There is a long discussion in the rebuttal letter but many of the points do not appear in the manuscript. In the manuscript, there is a major discussion on the correlation of chemical shift of Ca and the proposal of dual interaction and the complex is crucial for the reaction. However, in the rebuttal letter it is stated that “the formation of the supramolecular bonding complex does not mean that the reaction would occur” and “a major supramolecular complex does not mean that it would necessarily lead to a major product”. Instead of just explaining in the rebuttal letter, it is suggested to refine the discussion sections by adding the points such as “the complexation of catalyst and substrate is just a necessary process but often is not the rate-determining step”, “more favorable C-O bond cleavage determines the reaction outcome.”, etc.

Response: We appreciate the reviewer for further comments on our work. A related comment was added in the manuscript, see: “For this chalcogen bonding catalysis approach, the complexation of catalyst and ether is an essential process but is not the rate-determining step. Once the supramolecular bonding complex is formed, the more favorable C-O bond cleavage enabling the formation of a more stable carbocation determines the reaction outcome.”